# The Choice of the Most Appropriate Suture Threads for Pancreatic Anastomoses on the Basis of Their Mechanical Characteristics

**DOI:** 10.3390/biomedicines11041055

**Published:** 2023-03-30

**Authors:** Michele Pagnanelli, Francesco De Gaetano, Gennaro Nappo, Giovanni Capretti, Maria Laura Costantino, Alessandro Zerbi

**Affiliations:** 1Department of Biomedical Sciences, Humanitas University, Via Rita Levi Montalcini 4, 20072 Pieve Emanuele, Milan, Italy; 2IRCCS Humanitas Research Hospital, Via Manzoni 56, 20089 Rozzano, Milan, Italy; 3Department of Chemistry, Materials and Chemical Engineering “Giulio Natta”, Politecnico di Milano, 20133 Milano, Milan, Italy

**Keywords:** pancreaticoduodenectomy, pancreatic anastomoses, pancreatic fistula, suture material, Polydioxanone, Poliglecaprone 25

## Abstract

The choice of the most appropriate suture threads for pancreatic anastomoses may play an important role in reducing the incidence of post-operative pancreatic fistula (POPF). The literature on this topic is still not conclusive. The aim of this study was to analyze the mechanical characteristics of suture materials to find the best suture threads for pancreatic anastomoses. A single-axial electromagnetic actuation machine was used to obtain the stress–deformation relationship curves and to measure both the ultimate tensile strength (UTS) and the Young’s modulus at the 0–3% deformation range (E_0–3_) of four different suture materials (Poliglecaprone 25, Polydioxanone, Polyglactin 910, and Polypropylene) at baseline and after incubation in saline solution, bile, and pancreatic juice for 1, 3, and 7 days. Polydioxanone and Polypropylene showed stable values of UTS and E_0–3_ in all conditions. Polyglactin 910 presented significant UTS and E_0–3_ variations between different time intervals in all types of liquids analyzed. Poliglecaprone 25 lost half of its strength in all biological liquids analyzed but maintained low E_0–3_ values, which could reduce the risk of lacerations of soft tissues. These results suggest that Polydioxanone and Poliglecaprone 25 could be the best suture materials to use for pancreatic anastomoses. In vivo experiments will be organized to obtain further confirmations of this in vitro evidence.

## 1. Introduction

In pancreatic surgery, as usually occurs in other surgical specializations, the choice of the most adequate suture thread is related to the experience of the surgeons, the tradition of the school of surgery, or the resources available in the hospital. However, the use of the most adequate type of suture thread for each step of the interventions usually plays a fundamental role in the outcomes of the anastomoses and consequently the recovery of the patients.

After pancreatoduodenectomy, post-operative pancreatic fistula (POPF) is the most frequent and critical complication, ranging from 22% to 30% in high-volume specialized centers [1,2]. Severe POPF is responsible for the occurrence of other serious complications, including death, and has a wide impact also on the costs of hospitalization [3]. However, up to now, no surgical techniques [4,5,6,7,8,9,10]; position of drainages [11,12,13,14,15,16]; or use of drugs [17,18,19,20,21], stents [22,23,24,25,26,27], and surgical glues [28,29,30] have demonstrated a clear advantage in the reduction of POPF rate and its connected morbidity.

In the literature, there are few studies that analyze the suture threads after incubation in biological liquids, and none of these works have a close collaboration between the engineering knowledge and the surgical practical competence in this specific field. In the literature, the analysis of the characteristics of suture thread or other biological liquids was performed independently by engineering groups with low correlation with surgical practice [31,32], or by surgical teams [33,34,35,36,37,38,39], who pay more attention on the surgical outcomes of their results. On the contrary, the present work is based on a strong collaboration between engineering and surgery in order to obtain a deep correlation between the laboratory results on mechanical tests and the surgical outcomes in the operating room as well as in the post-operative course of patients undergoing pancreatic resections.

As a matter of fact, the first aim of this study was to analyze how bile and pancreatic juice could modify the mechanical characteristics of the suture materials commonly used in pancreatic surgery in order to find the most appropriate suture thread to realize a pancreatojejunostomy (PJ) with the lowest risk of POPF.

## 2. Materials and Methods

This project is part of a collaboration between the Department of Chemistry, Materials and Chemical Engineering “Giulio Natta” of Politecnico di Milano, Istituto Clinico Humanitas and Humanitas University, which enabled the creation of a multidisciplinary laboratory next to the hospital, aimed at promoting scientific innovation in surgery.

In the present study, we tested four suture materials among the most commonly used in pancreatic surgery: Poliglecaprone 25 (Ethicon Monocryl^TM^, Johnson&Johnson International, Diegem, Belgium), Polydioxanone (Ethicon PDS II^TM^, Johnson&Johnson International, Diegem, Belgium), Polyglactin 910 (Ethicon Vicryl^TM^, Johnson&Johnson International, Diegem, Belgium), and Polypropylene (Ethicon Prolene^TM^, Johnson&Johnson International, Diegem, Belgium). All the tested samples were 4-0 and 70 cm long.

Tensile tests were carried out through an Ultimate Tensile Machine (UTM) Materials Testing Systems—Synergie 200H (MTS System Corporation, Eden Prairie, MN, USA), which is a single-axial electromagnetic actuation machine equipped with a load cell of 100 N. To avoid misalignment of the suture threads during the test, two test grips were specifically designed. Standard test methods to characterize the properties of suture threads subjected to traction (ASTM D3822/D3822M-14 and ASTM D2256/D2256M-21) [40,41] were followed and integrated with the experiments presented in the literature [31,33,35,36] in order to create the test-specific protocol for this project.

According to this protocol, suture threads were fixed with sufficient effective distance between the grips in order to avoid the influence of the gripping system in the mechanical analysis of the wires. The grips were designed with the aim to obtain an ideal rupture of the samples at the middle of the wires, which means a perfect transmission of load through the sample, avoiding any stress concentration on the grips or the misalignment of the samples. Firstly, the gripping mechanism was 3D printed to validate the idea and verify its operation, and then they were manufactured by CNC technology in aluminum to be better performers over time to the applied load during the tests. In addition, to avoid metal-on-metal coupling and reduce the stresses applied on the wire when locked, Teflon tape was applied over the grip and around the screw head. Moreover, in order to reduce the risk of misalignments and other types of external stresses, the protocol included a system that aligned automatically when the samples were subjected to traction, by the presence of bearings in shafts, which allowed the free rotation of the grips when pulled, adding a further degree of freedom to the mechanism and allowing the alignment of the bottom and the upper grips.

For a complete mechanical characterization of the suture threads, all tensile tests were carried out until the rupture of the sample, and in case of slippage of the wire or rupture at the grip, the tests were neglected and repeated to guarantee the effectivity of the tests (Figure 1).

Six threads of each suture material were tested in order to show statistical relevance of the project. To evaluate the change of the mechanical characteristics in the physiological condition, the sutures were tested at baseline (dry) and in three different wet conditions: saline solution, bile, and pancreatic juice at three-time steps (1, 3, and 7 days) for each condition. A total of 240 tests were performed.

Bile was collected from an external transhepatic biliary drainage of a single patient suffering from obstructive jaundice due to a locally advanced pancreatic adenocarcinoma and awaiting endoscopic retrograde cholangiopancreatography. Pancreatic juice was gathered from an abdominal drainage of a single patient suffering from pancreatic fistula after pancreatoduodenectomy with PJ on isolated loop. After obtaining the patients informed consent, both fluids were collected and analyzed daily. Bile had an average pH level of 8.0, and pancreatic juice had a mean amylase level of 17,000 U/mL. Bacterial cultures were always negative for contamination.

Each sample was immersed in 14 mL sterile falcon filled with 10 mL of fluid and placed in an incubator at 37 °C. Fluids were replaced every day and tests were performed after 1, 3, and 7 days of continuous incubation. This timing was chosen in consideration of the average time of development of pancreatic or biliary fistula, assuming that, after seven days from the operation, the healing process was not influenced by the suture threads anymore.

The main outputs of the tests were the force in newtons (N) to determine the rupture of the suture, the elapsed time (s) to the breaking point, and the displacement of the actuator at the upper grip Δ*L* (mm) with respect to the initial condition. With this data, the values of the stress σ (MPa) and of the deformation *ε* (%) were obtained, assuming linear-elastic behavior of the material using the following equations: σ = F/$A_{0}$ and ε = ΔL/$L_{0}$ ($A_{0}$ = initial cross-section of the thread; $L_{0}$ = the initial length of the wire).

This enabled us to obtain a stress–deformation curve illustrating the ultimate tensile strength (UTS), which is the highest strength to break the suture thread while performing a suture knot, and the Young’s modulus at the 0–3% range of deformation (E_0–3_), which shows the stiffness of the suture thread when stretched at low intensities. The UTS was represented by the curve itself, while the E_0–3_ was calculated at such a low interval because it was expected that during and after the intervention, the suture threads were subjected to small deformations such as bowel loop edema or peristalsis (Figure 2). The threads showed a high non-linear elastic response: the Poliglecaprone 25 behavior seemed bi-linear, while the Polypropylene showed hyper-elastic behavior. However, in the deformation range considered (0–3%), all the threads showed a linear-elastic behavior.

### Statistical Analysis

The mean value of six identical measurements for each suture thread and each testing condition was compared to the baseline in order to evaluate the presence of statistically significant differences in terms of loss of UTS and E_0–3_ after incubation in different types of liquid analyzed. The Shapiro–Wilk test was used to evaluate the normality of the data. To find the statistically significant differences in our analysis, we used one-way analysis of variance (ANOVA) for data with normal distribution and Kruskal–Wallis tests for data without normal distribution. For data without normal distribution, the median was used instead of the mean. *p* values < 0.05 were considered statistically significant. Statistical analysis was performed with SPSS software (SPSS Inc. version 27 for Macintosh, IBM, Chicago, IL, USA).

## 3. Results

Each suture thread has specific mechanical characteristics, which are represented by peculiar stress–deformation curves with typical UTS and E_0–3_ values (Figure 3, Figure 4, Figure 5 and Figure 6, Table 1, Table 2, Table 3 and Table 4).

At baseline, Poliglecaprone 25 and Polydioxanone had similar UTS (Poliglecaprone 25 = 1721.1 MPa; Polydioxanone = 1726.3 MPa). This value was not far from the UTS of Polyglactin 910 (1664.5 MPa), which means that these two monofilament threads were created to have the same strength of a braided wires, even though they both had a very different level of stiffness expressed by their E_0–3_. As a matter of fact, Poliglecaprone 25 had the lowest E_0–3_ at baseline (789.3 MPa), while Polyglactin 910 was much stiffer, with E_0–3_ ten times higher than Poliglecaprone 25 (7663.9 MPa). Polydioxanone was more rigid than Poliglecaprone 25, but much less rigid than Polyglactin 910 (E_0–3_ = 2660.3 MPa). On the contrary, Polypropylene had the lowest UTS at baseline (904.2 MPa), being the most susceptible to rupture during elongation. However, it had the same stiffness of Polydioxanone (E_0–3_ = 2504.2 MPa), four times higher than Poliglecaprone 25 (Table 5).

The results of our tests enabled us to evaluate the changes in strength and stiffness of any suture thread after short and prolonged exposure to different biological liquids.

In the analysis of UTS changes, all suture materials showed statistically significant variations between the baseline and the wet conditions, regardless of the type of liquid analyzed. Interestingly, all of them, apart from Polypropylene, showed statistically significance UTS variations even after incubation in saline solution, while Polydioxanone did not reveal statistically significant UTS changes when treated in pancreatic juice. On the contrary, Poliglecaprone 25 and Polyglactin 910 also showed statistically significant UTS variations between different time intervals in any type of liquid analyzed. After 7 days of incubation, at the worst possible condition analyzed in the study, all suture threads showed statistically significant UTS variations, especially when treated in bile, but only Poliglecaprone 25 presented a reduction of almost half of its original UTS value, even when incubated in saline solution (1105.7 MPa in saline solution; 967.4 MPa in bile; 1050.9 MPa in pancreatic juice) (Figure 7, Table 5 and Table 6).

In the analysis of E_0–3_ changes, all suture threads showed statistically significant reductions between the dry and wet conditions. Only Polypropylene and Poliglecaprone 25 had statistically significant variations between different time intervals after incubation in bile. After 7 days of incubation, the worst condition analyzed in the study, Polydioxanone maintained almost stable levels of E_0–3_ in all conditions (2404.2 MPa at baseline; 2045.5 MPa in bile; 2039.8 MPa in pancreatic juice) as well as Polypropylene with a similar stiffness (2660.4 MPa at baseline; 2636.5 MPa in bile; 2214.3 MPa in pancreatic juice). Poliglecaprone 25 maintained the lowest level of E_0–3_, even in bile (571.6 MPa) or pancreatic juice (568.0), which means it is considerably more deformable than other suture threads when subjected to traction. Conversely, Polyglactin 910 was confirmed to be the stiffest suture material in this group, even after incubation in bile (6120.2 MPa) or pancreatic juice (6680.1 MPa) (Figure 8, Table 5 and Table 7).

## 4. Discussion

The incidence of POPF primarily influences the postoperative course of patients undergoing pancreatoduodenectomy in terms of morbidity and mortality [42], access to oncological treatments [43], and the level of healthcare costs [3]. POPF is a complex process influenced by a wide number of different factors such as the pancreatic texture, the diameter of the main pancreatic duct, the type of pancreatic pathology requiring resection, and the amount of intraoperative blood loss [44]. However, the type of suture threads used to perform pancreatic anastomoses may influence the outcome of the strength and resistance of pancreatic sutures, and consequently this may play an important role in the incidence of POPF and its grade. In the last decades, many different surgical techniques have been evaluated to contain the development of POPF [4,5,6,7,8,9,10], but only few studies have analyzed the mechanical characteristics of suture threads after incubation in bile and pancreatic juice as well as their Young’s modulus with the aim of finding the most appropriate suture threads for pancreatic anastomoses [45]. On the contrary, the present study is based on a multidisciplinary collaboration between engineering and surgery with the aim to obtain more solid conclusions in such a wide experimental field.

In our work, all tests were performed with a single-axial electromagnetic Ultimate Tensile Machine (UTM)—Synergie 200H (MTS System Corporation, Eden Prairie, MN, USA) equipped with a load cell of 100N. The groups of Naleway et al., Karaman et al., and Gierek et al. used similar machines and paid attention to settle the suture threads in order to let it break in the middle rather than next to the anchorage knot [31,35,45]. However, their system does not have a grip mechanism able to self-align the suture threads during the test, avoiding any misalignments and consequently errors during the acquisition of the stress–deformation curves used to evaluate the UTS and Young’s modulus. On the contrary, all other studies in literature used a much simpler tensiometer with a shorter length of the suture threads analyzed, which is easier to use, but rather less precise in the measurement.

In the present work, suture threads were analyzed at baseline, on days 1, 3, and 7, after incubation, since the first post-operative week is tendentially the period in which POPF develops and becomes evident. Hence, the first 14 days of the healing process of a surgical wound/anastomosis consist of the exudative (0–4 days) and the proliferative (4–14 days) phases [46]. This means that suture threads play the main role in the scarring process of anastomosis in the first two weeks after the intervention and even more in the first seven days. After this period, if the POPF has developed, no suture thread can solve or reduce it. However, most of the studies in the literature, apart from the ones conducted by Muftuoglu et al. and Karaman et al., analyzed longer periods of incubation, which provide more information about the alteration of surgical threads, but are less applicable to real surgical outcomes [34,35].

We decided to analyze Polypropylene, Polydioxanone, and Polyglactin 910 because they are the most commonly used suture threads in pancreatic surgery [47]. Furthermore, differently from other studies in the literature [33,34,35], we also analyzed Poliglecaprone 25 because it is an absorbable monofilament with a very high level of deformability and is widely used in pancreatic surgery worldwide [47]. This choice enabled us to have a comprehensive evaluation of the mechanical characteristics of suture martials with different physical features at baseline (monofilament/braided—absorbable/non-absorbable). The paper of Gierek et al. is one of the few in the literature in which Poliglecaprone 25 was analyzed with Polydioxanone and Polyglactin 910. The authors performed incubation tests in five different environments (physiological saline, sterile and contaminated pancreatic juice, sterile and contaminated bile) for 7, 14, 21, and 28 days and demonstrated that contaminated environments seem to influence the level of degradation of suture materials, even though the antibacterial coating of some of these suture threads does not modify the resistance of sutures themselves. Moreover, as in our study, their results showed that Polyglactin 910 had a significant degradation after incubation, especially in bile, while Polydioxanone had the highest resistance and the longest alteration time in all the analyzed biological liquids [45].

Freudenberg et al. obtained similar results by incubating nine synthetic absorbable suture threads in blood and different gastrointestinal fluids for 7, 14, and 21 days. They confirmed the tendency to degradation of Polyglactin 910 in bile and the stability of Polydioxanone after incubation in pancreatic juice or bile. They also demonstrated the low resistance of Polydioxanone in an acid environment and the stability of Poliglecaprone 25 in pancreatic juice [39]. Similar conclusions were reached by Muftuoglu et al. and Karaman et al. Both the studies demonstrated that Polyglactin 910 has the highest UTS at baseline, but it loses most of its strength after incubation in biological liquids. Moreover, both Polypropylene and Polydioxanone retain most of their initial strength after incubation, even though Polydioxanone is more indicated than Polypropylene for PJ due to its absorbable nature [34,35].

On the contrary, although presenting similar results about Polypropylene, Polydioxanone, and Polyglactin 910, Andrianello et al. declared the superiority of Polyester for PJ. In their work, they assumed that Polyester is easier and less traumatic to knot as well as being more resistant to the degradation prompted by the pancreatic juice [33]. Nevertheless, since Polyester is a braided and non-absorbable material, it is much stiffer and more prone to induce inflammatory reactions than Polydioxanone and Poliglecaprone 25, with higher risk of anastomotic stenosis, tissue lacerations, and stump pancreatitis. For this reason, we decided to exclude Polyester from our analysis, which should not be considered as a limitation of the present study.

In our study, we decided to analyze both UTS and E_0–3_ of suture threads after incubation in biological liquids since we believe that the measurement of just the strength to the breaking point, which in our case coincide always with UTS (Figure 2), would not be enough to evaluate the best suture material in pancreatic surgery. As a matter of fact, while UTS provides important information about the resistance of the suture material when knotted, stretched, or exposed to biological fluids, the E_0–3_ shows the stiffness of the suture thread that should be compared to the rigidity of the organic tissues during and after the intervention. According to our results, Polydioxanone presented statistically significant differences only in the comparison between the baseline (dry) and the wet condition, regardless the type of liquid analyzed (saline solution, bile, or pancreatic juice). Moreover, it maintained stability in terms of UTS and E_0–3_ in all biological liquids tested, especially in pancreatic juice (Figure 4). Similar outcomes were obtained with Polypropylene, even though it had half the value of UTS and almost the same level of E_0–3_ compared to Polydioxanone (Figure 6). Considering this stability after exposure to biological liquids, Polypropylene could potentially be a good choice for the inner layer of the PJ. However, due to its non-absorbable nature, which makes it more prone to induce inflammation, anastomotic stenosis, or foreign body reactions, Polypropylene should not be preferred to Polydioxanone, as confirmed in other studies in the literature [34,35,47].

Conversely, Polyglactin 910 showed statistically significance reductions of UTS and E_0–3_ in both bile and pancreatic juice, as well as in the comparison between different time steps (Figure 5). This tendency justifies the results of several studies in the literature that demonstrated the complete degradation of Polyglactin 910 after longer periods of incubation and prevented its use for PJ [33,37,39]. In addition, being a braided material, the E_0–3_ of Polyglactin 910 is considerably higher than other suture threads, which makes it less adequate, even for the outer layer of PJ, due to the risk of lacerating tissues and increasing the incidence of POPF.

On the other hand, Poliglecaprone 25 has a considerable reduction of its UTS after incubation in bile and pancreatic juice but maintains low levels of E_0–3_ at all conditions (Figure 3). This means that Poliglecaprone 25 tends to lose strength after prolonged exposure to bile and pancreatic juice, but it is more deformable than other suture thread to adapt to physiological modification of human tissues (pancreatic consistency, bowel movements, edema) without tearing them. Moreover, the UTS value for all the suture threads measured in this study and analyzed in the literature are related to thread deformation values (40% to 70%) that are incompatible with the surgical application for which they are used, providing even more importance to the use of Young’s modulus at low deformation (E_0–3_) for comparison between threads.

Although our tests clearly show the effect of bile and pancreatic juice on the strength and stiffness of different suture threads, the mechanism responsible for these outcomes is still difficult to understand. Indeed, bile has no active enzymes, while pancreatic juice has proteolytic, lipolytic, and amylolytic proenzymes as well as active lipase and amylase, but none of the suture threads analyzed in our study contain proteins that could suffer from proteolytic degradation [34]. On the other hand, Polyglactin 910, Poliglecaprone 25, and Polydioxanone, being absorbable materials, may be altered by hydrolysis, but this process alone could not explain the entire effects on suture threads. The same is the case for the pH of biological liquids, which seem to play an important role in this degradation mechanism. As demonstrated by Tomihata et al., Polyglactin 910 and Poliglecaprone 25 suffer from an alkaline environment, while Polydioxanone loses strength when incubated in acid solutions [32]. Moreover, the works of Gierek et al. and Chung et al. demonstrated the influence of contaminated environments on degradation of sutures, and the former group also identified the physical effects of their damages on suture materials through an electric microscope [45,48]. These results demonstrate that the degradation of suture threads is a multifactorial process, and it is highly dependent on the specific characteristics of different biological liquids they are exposed to.

As a consequence, it goes without saying that the choice of the most appropriate suture material for any step of the pancreatic intervention should consider not only the resistance of suture materials but also their action on biological tissues. Hence, the inner layer of the PJ seems to require an absorbable suture thread with high resistance to pancreatic juice and a medium level of stiffness to adapt to the intestinal mucosa without lacerating it but keeping the two sides of the anastomosis enfronted. On the contrary, the outer layer of the PJ would benefit from an absorbable material with lower strength and higher elasticity, which would be able to fix the intestinal loop to the pancreatic capsule without excessive tractions that would generate lacerations. Therefore, we consider Polydioxanone as the best suture material for the inner layer of PJ and Poliglecaprone 25 for the outer layer of PJ.

The in vitro results of this work also showed good clinical outcomes in our surgical practice, and we aim to obtain further confirmations through animal experiments that are going to be performed in the near future, in order to better understand how the other mechanisms involved in the healing process in pancreatic anastomoses could influence the mechanical characteristics of suture threads. Moreover, this study is part of a wider collaborative project with the Engineering Faculty of Politecnico di Milano aimed at characterizing the pancreatic tissue with a bio-nanoindenter in order to realize a realistic phantom of the pancreas. Although this analysis is still in fieri, the first draft data enabled us to understand important characteristics of the pancreatic tissue, especially in terms of Young’s modulus, which seemed to confirm our conclusions regarding the adequate use of suture threads.

## 5. Conclusions

The mechanism of development of the POPF is multifactorial and difficult to prevent or contrast. Most of the time, the size of the pancreatic stump, the consistency of its parenchyma, and the small width of the main pancreatic duct are the major predisposing factors in this process. However, the right choice of the suture threads for PJ may play an important role in order to reduce the incidence of POPF and its consequences.

The present study, with the evaluation of the UTS and E_0–3_ of suture threads after incubation in bile and pancreatic juice for 1 to 7 days, enabled us to show suture threads reaction when exposed to biological liquids and, consequently, their possible role in pancreatic anastomoses. On the one hand, Polydioxanone, maintaining stable UTS and E_0–3_ values in all the analyzed conditions, seemed to be the best suture thread for the inner layer of PJ, between the main pancreatic duct and intestinal mucosa in direct contact with pancreatic juice. On the other hand, Poliglecaprone 25, thanks to its low Young’s modulus and consequently its low tendency to cause soft tissue lacerations, seemed to be the best suture material for the outer layer of PJ, between the pancreatic capsule and the intestinal loop. This work confirms and supports, from an engineering point of view, a surgical habit already in use in many high-volume pancreatic units. In vivo experiments on animals will be organized as integrative outcomes of our project in order to obtain further confirmations of the in vitro evidence we obtained with our experiments.

## Figures and Tables

**Figure 1 biomedicines-11-01055-f001:**
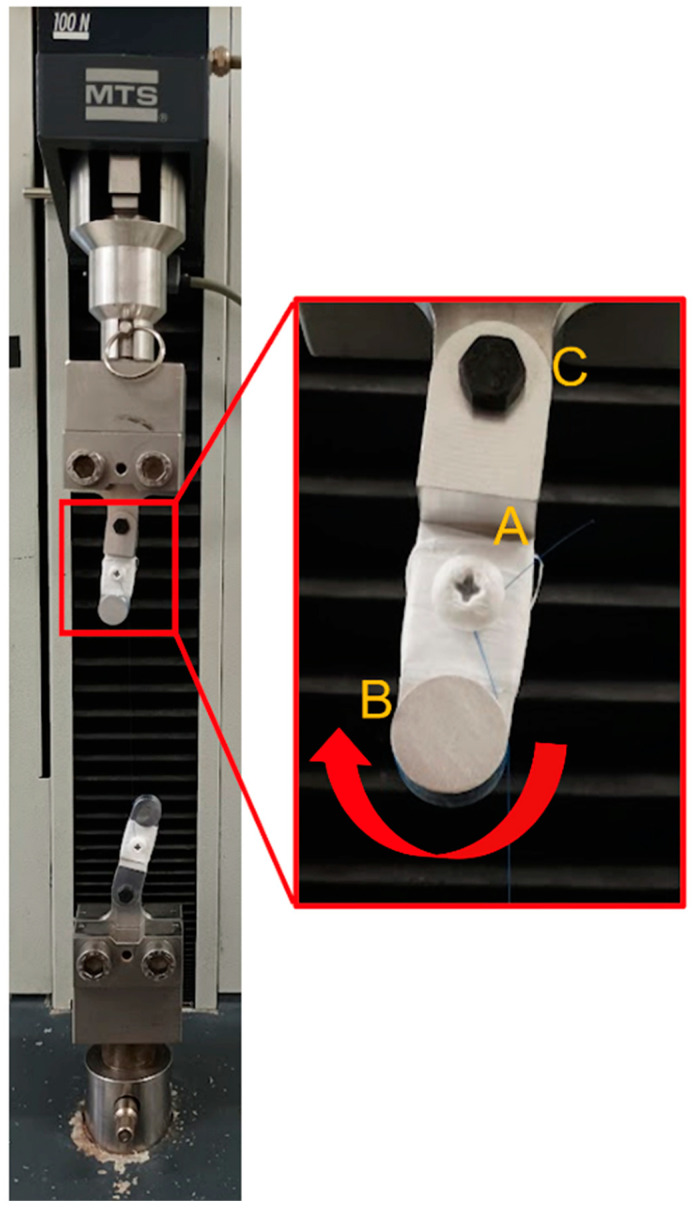
Details of the designed grips. The threads are mounted on the grips as described: a surgical knot is done around the screw (**A**), and the suture thread is passed one time around the cylinder (**B**). Finally, the grips are separated until the suture thread is tight. The right self-alignment of the thread is guaranteed by the bearing placed in (**C**) allowing free rotation of the grip.

**Figure 2 biomedicines-11-01055-f002:**
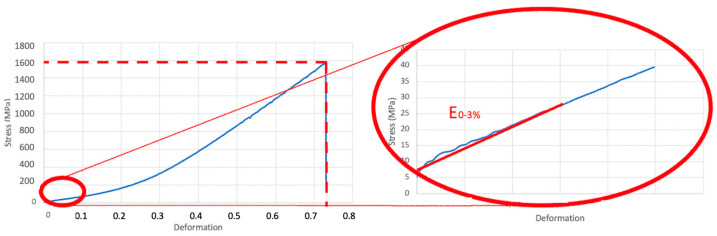
Stress–deformation curve with ultimate tensile strength (UTS) and Young’s modulus at the 0–3% range of deformation (E_0–3_).

**Figure 3 biomedicines-11-01055-f003:**
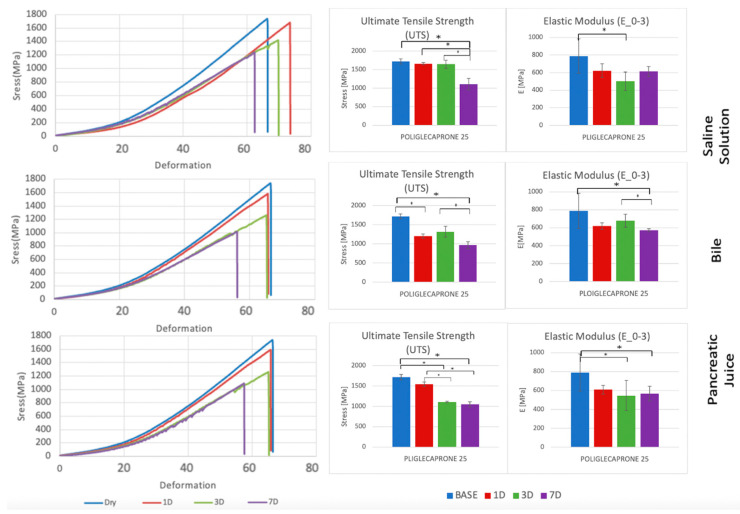
Comparison of stress–deformation curves of Poliglecaprone 25 incubated in saline solution, bile, and pancreatic juice at different time steps (**left**). Corresponding UTS and E_0–3_ (**right**). * *p* < 0.05.

**Figure 4 biomedicines-11-01055-f004:**
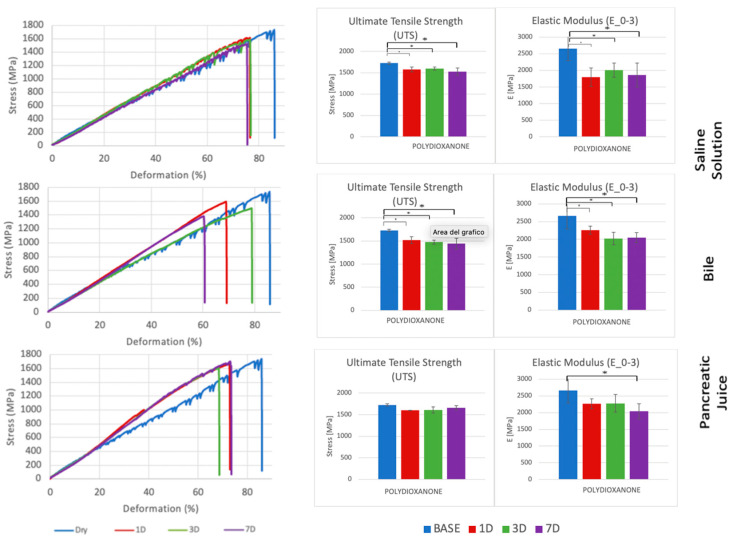
Comparison of stress–deformation curves of Polydioxanone incubated in saline solution, bile, and pancreatic juice at different time steps (**left**). Corresponding UTS and E_0–3_ (**right**). * *p* < 0.05.

**Figure 5 biomedicines-11-01055-f005:**
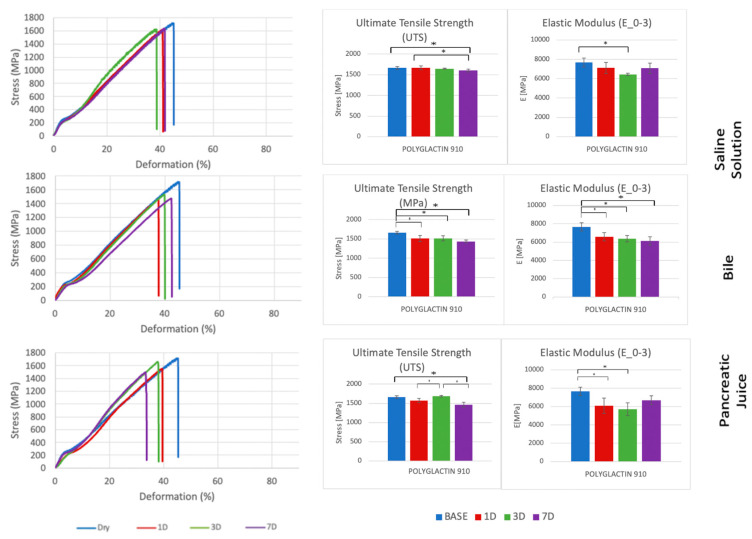
Comparison of stress-deformation curves of Polyglactin 910 incubated in saline solution, bile, and pancreatic juice at different time steps (**left**). Corresponding UTS and E_0–3_ (**right**). * *p* < 0.05.

**Figure 6 biomedicines-11-01055-f006:**
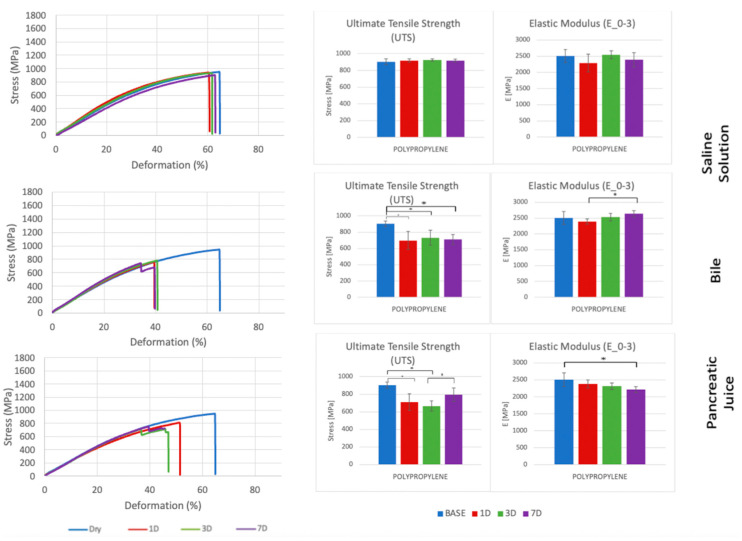
Comparison of stress–deformation curves of Polypropylene incubated in saline solution, bile, and pancreatic juice at different time steps (**left**). Corresponding UTS and E_0–3_ (**right**). * *p* < 0.05.

**Figure 7 biomedicines-11-01055-f007:**
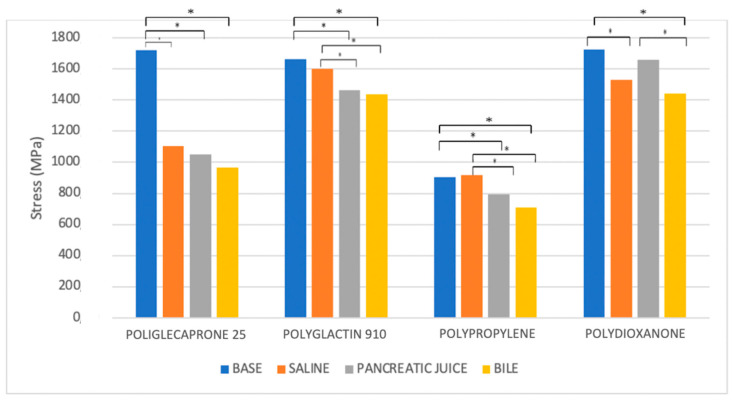
Comparison of the ultimate tensile strength (UTS) of different suture threads after 7 days of incubation in saline solution, bile, and pancreatic juice. * *p* < 0.05.

**Figure 8 biomedicines-11-01055-f008:**
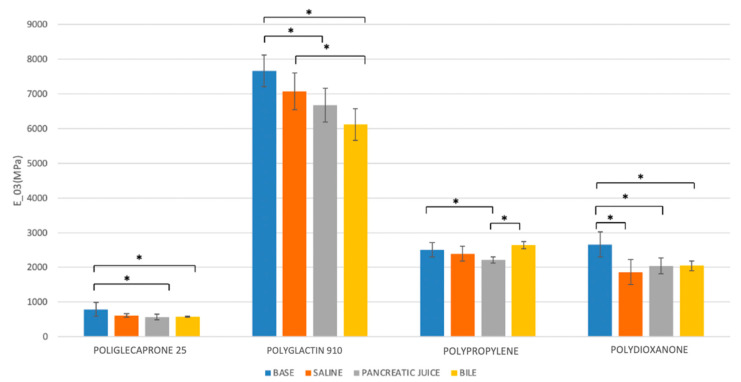
Comparison of the Young’s modulus at the 0–3% range of deformation (E_0–3_) of different suture threads after 7 days of incubation in saline solution, bile, and pancreatic juice. * *p* < 0.05.

**Table 1 biomedicines-11-01055-t001:** UTS and E_0–3_ of Poliglecaprone 25 incubated in saline solution, bile, and pancreatic juice at different time steps.

UTS [MPa]—Poliglecaprone 25			
	Saline Solution	Bile	Pancreatic Juice
**Time intervals**	*p*	*p*	*p*
**1 VS. 0**	0.102	**0.013**	0.221
**3 VS. 0**	0.153	0.072	**0.001**
**3 VS. 1**	0.838	0.488	**0.034**
**7 VS. 0**	**0.001**	**0.001**	**0.001**
**7 VS. 1**	**0.020**	0.060	**0.011**
**7 VS. 3**	**0.011**	**0.010**	0.683
**E_0–3_ [MPa]—Poliglecaprone 25**			
	**Saline Solution**	**Bile**	**Pancreatic Juice**
**Time intervals**	*p*	*p*	*p*
**1 VS. 0**	0.165	0.094	0.079
**3 VS. 0**	**0.004**	0.488	**0.045**
**3 VS. 1**	0.131	0.327	0.806
**7 VS. 0**	0.111	**0.001**	**0.022**
**7 VS. 1**	0.838	0.131	0.596
**7 VS. 3**	0.191	**0.013**	0.775

**Table 2 biomedicines-11-01055-t002:** UTS and E_0–3_ of Polydioxanone incubated in saline solution, bile, and pancreatic juice at different time steps.

UTS [MPa]—Polydioxanone			
	Saline Solution	Bile	Pancreatic Juice
**Time intervals**	*p*	*p*	*p*
**1 VS. 0**	**0.002**	**0.001**	0.080
**3 VS. 0**	**0.003**	**0.001**	0.108
**3 VS. 1**	0.998	0.819	0.999
**7 VS. 0**	**0.001**	**0.001**	0.539
**7 VS. 1**	0.516	0.392	0.631
**7 VS. 3**	0.424	0.878	0.722
**E_0–3_ [MPa]—Polydioxanone**			
	**Saline Solution**	**Bile**	**Pancreatic Juice**
**Time intervals**	*p*	*p*	*p*
**1 VS. 0**	**0.001**	**0.043**	0.113
**3 VS. 0**	**0.015**	**0.001**	0.127
**3 VS. 1**	0.708	0.341	1.00
**7 VS. 0**	**0.003**	**0.001**	**0.006**
**7 VS. 1**	0.988	0.438	0.539
**7 VS. 3**	0.876	0.998	0.500

**Table 3 biomedicines-11-01055-t003:** UTS and E_0–3_ of Polyglactin 910 incubated in saline solution, bile, and pancreatic juice at different time steps.

UTS [MPa]—Polyglactin 910			
	Saline Solution	Bile	Pancreatic Juice
**Time intervals**	*p*	*p*	*p*
**1 VS. 0**	0.903	**0.008**	0.111
**3 VS. 0**	0.514	**0.027**	0.514
**3 VS. 1**	0.596	0.653	**0.025**
**7 VS. 0**	**0.020**	**0.001**	**0.002**
**7 VS. 1**	**0.027**	0.327	0.121
**7 VS. 3**	0.094	0.153	**0.001**
**E_0–3_ [MPa]—Polyglactin 910**			
	**Saline Solution**	**Bile**	**Pancreatic Juice**
**Time intervals**	*p*	*p*	*p*
**1 VS. 0**	0.252	**0.030**	**0.005**
**3 VS. 0**	**0.002**	**0.003**	**0.001**
**3 VS. 1**	0.135	0.438	0.462
**7 VS. 0**	0.210	**0.001**	0.072
**7 VS. 1**	0.999	0.165	0.307
**7 VS. 3**	0.165	0.540	0.079

**Table 4 biomedicines-11-01055-t004:** UTS and E_0–3_ of Polypropylene incubated in saline solution, bile, and pancreatic juice at different time steps.

UTS [MPa]—Polypropylene			
	Saline Solution	Bile	Pancreatic Juice
**Time intervals**	*p*	*p*	*p*
**1 VS. 0**	0.781	**0.004**	**0.002**
**3 VS. 0**	0.376	**0.016**	**0.001**
**3 VS. 1**	0.895	0.913	0.759
**7 VS. 0**	0.793	**0.006**	0.093
**7 VS. 1**	1.000	0.996	0.268
**7 VS. 3**	0.885	0.973	**0.044**
**E_0–3_ [MPa]—Polypropylene**			
	**Saline Solution**	**Bile**	**Pancreatic Juice**
**Time intervals**	*p*	*p*	*p*
**1 VS. 0**	0.403	0.552	0.494
**3 VS. 0**	0.989	0.990	0.142
**3 VS. 1**	0.255	0.382	0.842
**7 VS. 0**	0.846	0.427	**0.012**
**7 VS. 1**	0.862	**0.042**	0.216
**7 VS. 3**	0.676	0.603	0.640

**Table 5 biomedicines-11-01055-t005:** Tensile test results of different suture threads at baseline and after incubation in saline solution, bile, and pancreatic juice.

Saline solution								
	Poliglecaprone 25 4-0		Polydioxanone 4-0		Polyglactin 910 4-0		Polypropylene 4-0	
Time	UTS [MPa]	E_0–3_ [MPa]	UTS [MPa]	E_0–3_ [MPa]	UTS [MPa]	E_0–3_ [MPa]	UTS [MPa]	E_0–3_ [MPa]
**Dry**	1721.1	789.3	1726.3	2660.3	1664.5	7663.9	904.2	2504.2
**1 Day**	1662.2	622.5	1578.4	1796.2	1667.4	7108.1	917.9	2289.5
**3 Days**	1646.6	503.9	1598.3	2006.1	1647.4	6445.8	924.9	2546.3
**7 Days**	1105.7	612.5	1529.9	1860.0	1601.9	7075.0	917.5	2394.5
**Bile**								
	**Poliglecaprone 25** **4-0**		**Polydioxanone** **4-0**		**Polyglactin 910** **4-0**		**Polypropylene** **4-0**	
**Time**	**UTS** **[MPa]**	**E_0–3_** **[MPa]**	**UTS** **[MPa]**	**E_0–3_** **[MPa]**	**UTS** **[MPa]**	**E_0–3_** **[MPa]**	**UTS** **[MPa]**	**E_0–3_** **[MPa]**
**Dry**	1721.1	789.3	1726.3	2660.3	1664.5	7663.9	904.2	2504.2
**1 Day**	1205.4	619.3	1517.8	2259.6	1512.8	6594.7	698.6	2390.4
**3 Days**	1321.5	679.1	1477.0	2019.4	1515.4	6363.9	732.1	2529.8
**7 Days**	967.4	571.6	1442.2	2045.5	1435.1	6120.2	710.1	2636.4
**Pancreatic Juice**								
	**Poliglecaprone 25** **4-0**		**Polydioxanone** **4-0**		**Polyglactin 910** **4-0**		**Polypropylene** **4-0**	
**Time**	**UTS** **[MPa]**	**E_0–3_** **[MPa]**	**UTS** **[MPa]**	**E_0–3_** **[MPa]**	**UTS** **[MPa]**	**E_0–3_** **[MPa]**	**UTS** **[MPa]**	**E_0–3_** **[MPa]**
**Dry**	1721.1	789.3	1726.3	2660.3	1664.5	7663.9	904.2	2504.2
**1 Day**	1547.1	608.6	1599.1	2265.0	1573.7	6089.8	710.5	2383.7
**3 Days**	1108.0	546.1	1606.8	2275.9	1685.6	5711.1	666.8	2314.3
**7 Days**	1050.9	568.0	1658.9	2039.8	1461.7	6680.1	793.9	2214.3

**Table 6 biomedicines-11-01055-t006:** Comparison of the UTS of different suture threads after 7 days of incubation in saline solution, bile, and pancreatic juice.

UTS [MPa] after 7 days				
	Poliglecaprone 25	Polyglactin 910	Polypropylene	Polydioxanone
*Fluids* *Comparison*	*p*	*p*	*p*	*p*
**NaCl VS.** **Dry**	**0.027**	0.191	0.979	**0.002**
**Bile VS.** **Dry**	**0.001**	**0.001**	**0.019**	**0.001**
**Pancreas VS.** **Dry**	**0.004**	**0.001**	**0.001**	0.221
**Bile VS.** **NaCl**	0.121	**0.014**	**0.001**	0.514
**Pancreas VS.** **NaCl**	0.514	**0.050**	**0.008**	0.066
**Bile VS.** **Pancreas**	0.369	0.624	0.096	**0.013**

**Table 7 biomedicines-11-01055-t007:** Comparison of the E_0–3_ of different suture threads after 7 days of incubation in saline solution, bile, and pancreatic juice.

E_0–3_ [MPa]—after 7 days				
	Poliglecaprone 25	Polyglactin 910	Polypropylene	Polydioxanone
*Fluids* *Comparison*	*p*	*p*	*p*	*p*
**NaCl VS.** **Dry**	0.082	0.250	0.714	**0.001**
**Bile VS.** **Dry**	**0.024**	**0.001**	0.584	**0.015**
**Pancreas VS.** **Dry**	**0.022**	**0.021**	**0.049**	**0.014**
**Bile VS.** **NaCl**	0.846	**0.025**	0.120	0.741
**Pancreas VS.** **NaCl**	0.917	0.579	0.326	0.758
**Bile VS.** **Pancreas**	1.000	0.289	**0.003**	1.000

## Data Availability

All results are presented in this article.

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
