# Peer review of "The Choice of the Most Appropriate Suture Threads for Pancreatic Anastomoses on the Basis of Their Mechanical Characteristics"

_biomedicines, 2023, doi:10.3390/biomedicines11041055_

Round 1

Reviewer 1 Report

Major concern: The authors presumed that the mechanical characters plays an vital role in POPF, but there is no experimental prove in this paper, since POPF is a very complex process, the authors should provide evidence that the suture is critical to the POPF. The authors absolutely need an animal experiment!

Minor: They in vitro saline, bile and pancreatic juice are not the only factors contributed to the degradation of the suture, there is inflammation, strength of the suture knot, et al. The authors should pay attention to this. 

Otherwise, this is just a research on the different sutures resistant to the bile and pancreatic juice. 

Author Response

I would like to thank the reviewer for the insightful and very constructive comments and suggestions, that helped us to improve the quality of our manuscript.

Major concern: Thank you for your very interesting comments and considerations. I agree that POPF is a complex process, and that suture is only one of the involved factors: we added a comment on this point in the discussion.  I also agree that animal experiments would add high strength to our study, and we have planned to perform them in the nearby future, to confirm the data presented in the article. Our study is in fact part of a wider project in collaboration with engineers aimed to characterize the pancreatic tissue with a bio-nanoindenter to realize a realistic phantom of the pancreas. Although this analysis is still in fieri, the first draft data enabled us to understand important characteristics of the pancreatic tissue especially in terms of Young’s modulus, which seemed to confirm our conclusions regarding the adequate use of suture threads. We added a comment on these aspects in the discussion.

Minor concern: This is an excellent point too. Thank you for having underlined it.  We have added a comment on this aspect in the discussion.

Reviewer 2 Report

The paper concerns an experimental analysis for assessing among four different suture threads which is the more suitable for pancreatic anastomoses. To this aim, the tensile strength and Young’s modulus for low deformations are determined by tensile tests; moreover, the effects on the above mechanical parameters of 1, 3 and 7 days of incubation in saline solution, bile and pancreatic juice are evaluated. Tensile tests are performed by using ad-hoc designed and realized grips.

The subject of the paper is interesting, as well as the presented experimental results. Anyway, in my opinion, there are major issues to be solved before accepting the paper for publication:

1) Section 2, line 118: no assumptions of a linear elastic behavior are needed for relate the normal stress and the longitudinal strain to the tensile force and to the elongation, respectively. Consider also that the threads show in the experiments an evident non-linear elastic response (see Figures 3-6).

2) The paper uses the terms “pancreatic anastomoses”, “pancreatoduodenectomy”, and “pancreatojejunostomy”: please, clarify if those terms indicate the same surgical operation, otherwise check the right use in the text.

3) Even if for a thread the term “elastic modulus” can be accepted, since only longitudinal deformations are meaningful, I think that the term “Young’s modulus” is more appropriate.

4) Figure 1: there is a repeated sentence in the caption.

5) Figures 3-6: the curves show different kinds of curvatures. This aspect is worth to be commented for studying the performances of each material as a suture thread.

6) Figures 3-6: please, improve the quality of the diagrams on the right side.

7) Section 2.1, lines 311-313: this sentence substantially repeats what was already said in some sentences above.

8) Figures 7-8: please use for the threads the same name used elsewhere

9) Pages 12-13, from line 380: the text appears to be not related to the text above. Moreover, the text in lines 380-456 appears pertinent for the Introduction (and, actually, some concepts are repeated from the Introduction), not for the part devoted to the discussion of results.

10) Pages 13-14, lines 457-534: the conclusions seem to be not sufficiently supported by the discussion of the results: please, rewrite this part clearly presenting the expected performances of suture threads to be employed in each part of the PJ (possibly, also in the light of data about mechanical parameters of the involved organs), and explaining more clearly why one kind of thread is more suitable than the others.

11) The paper goes from Section 2.1 directly to Section 5: probably something is missed. Please, revise.

Author Response

I would like to thank the reviewer for the insightful and very constructive comments and suggestions, that helped us to improve the quality of our manuscript. Shown below the point-to-point responses:

  1. I appreciated for this consideration. I changed the sentence as follow: “With this data, the value of the stress σ [MPa] and of the deformation ε [%] were obtained using the following equations: σ = F/A_0 and ε = ΔL/L_0 (A_0 = initial cross-section of the thread; L_0 = the initial length of the wire)”. The threads show a high non-linear elastic response: the Poliglecaprone 25 behavior (fig 3) seems bi-linear while the Polypropylene (fig. 6) shows hyperplastic behavior. However, in the deformation range considered (0-3%) all the threads show a linear-elastic behavior.
  2. Our tests were conducted on suture threads commonly used in pancreatic anastomoses performed during pancreatoduodenectomy interventions. The most common reconstruction technique is through a pancreatojejunostomy, so I specified this specific type of anastomosis. I modified the text to make the concept clearer.
  3. For a tensile test thread, the term " elastic modulus " is accepted in several papers to describe the behavior of the tested threads, but I agree with the reviewer that " Young’s modulus " is more appropriate.
  4. I am sorry for the mistake. I modified that as suggested.
  5. I thank the reviewer for this correct request since the curve behavior of the threads is obviously different. However, these tests were carried out to understand the clinical implications of this behavior. This is the reason why I decided to make all the comparisons in the range between 0 and 3% of deformation. In this range of deformation, all the threads have the same behavior and therefore I decided not to study the behavior when the deformation is higher.
  6. I modified the Figures as suggested.
  7. I appreciated the suggestion of the reviewer. I modified the text as suggested.
  8. I made the modification as suggested.
  9. I appreciated the suggestion of the reviewer. I modified the text as suggested.
  10. I appreciated the suggestion of the reviewer. I integrated the conclusions as suggested.
  11. I am sorry for the mistake. I modified this as suggested.

Round 2

Reviewer 1 Report

still suggest animal study, otherwise delete "pancreatic anastomoses " in the title. 

Author Response

Dear reviewer, thank you for your insightful and very constructive suggestion.

I revised the paper in the abstract and the conclusions underlining that in-vivo experiments on animals will be organized as integrative aims of our project to obtain further confirmations of the in-vitro evidence we got with our experiments.

Reviewer 2 Report

The Authors revised the paper satisfactorily.

Author Response

Dear reviewer, thank you very much for your comments.